

# Comparison of global UV irradiance measurements between a BTS CCD-array and a Brewer spectroradiometers

Carmen González[1,2], José M. Vilaplana[1], José A. Bogeat[3], Antonio Serrano[2]

[1]Departamento de Observación de la Tierra y la Atmósfera, Instituto Nacional de Técnica Aeroespacial (INTA), El Arenosillo, Huelva, España
[2]Departamento de Física, Instituto del Agua, Cambio Climático y Sostenibilidad, Facultad de Ciencias, Universidad de Extremadura, Badajoz, España
[3]Centro de Experimentación de El Arenosillo (CEDEA), Instituto Nacional de Técnica Aeroespacial (INTA), El Arenosillo, Huelva, España

*Correspondence to*: Carmen González (cgonher@inta.es)

**Abstract.** Spectral measurements of UV irradiance are of great importance to ensure human health protection as well as to support scientific research. To perform these measurements, double monochromator scanning spectroradiometers are the preferred devices, thanks to their linearity and stray-light reduction. However, because of their high cost and demanding maintenance, CCD-array-based spectroradiometers are increasingly used for monitoring UV irradiance. Nevertheless, CCD-array spectroradiometers have specific limitations, such as a high detection threshold or stray-light contamination. To overcome these challenges, several manufacturers are striving to develop improved instrumentation. In particular, Gigahertz-Optik GmbH has developed the stray-light-reduced BTS2048-UV-S spectroradiometer series (from now on called BTS). In this study, the long-term performance of the BTS and its seasonal behavior, regarding global UV irradiance, has been assessed. To carry out the analysis, BTS' irradiance measurements have been compared against measurements of the Brewer MK-III #150 scanning spectrophotometer during three campaigns. A total of 711 simultaneous spectra, measured under cloud-free conditions and covering a wide range of solar zenith angle (from 14° to 70°) and UV index (from 2.4 to 10.6), are used for the comparison. During the three measurement campaigns, the global UV spectral ratio BTS/Brewer was almost constant (at around 0.93) in the 300–360 nm region for solar zenith angles (SZAs) below 70°. Thus, the BTS calibration was stable during the whole period of study (~1.5 years). Likewise, it showed no seasonal nor SZA significant dependence in this wavelength region. Regarding the UV index, a good correlation between the BTS and the Brewer #150 was found, i.e. the dynamic range of the BTS is comparable to that of the Brewer #150. These results confirm the quality of the long-term performance of the BTS array spectroradiometer to measure global UV irradiance.

## 1 Introduction

Prolonged exposure to solar UV radiation has adverse effects on the eye, immune system and skin of both humans (Cullen et al., 1984; Armstrong and Kricker, 1993; Garssen et al., 1996) and animals (Kripke, 1974; Doughty and Cullen, 1990; Eller et al., 1994) given that UV photons may damage DNA (deoxyribonucleic acid), proteins and lipids (Beukers and Berends,





1960; Häder and Brodhun, 1991; Ogura et al., 1991). Moreover, this radiation can also be harmful to materials (Lawrence and Weir, 1973; Hon and Chang, 1984; Capjack et al., 1994; Andrady et al., 2019) and a great deal of species such as forests (Sullivan and Teramura, 1988; Musil and Wand, 1993), phytoplankton (Smith et al., 1980; Döhler and Biermann, 1987;

Ekelund, 1990) and crops (Caldwell, 1968; Teramura, 1980; Krupa and Kickert, 1989). Spectral measurements are needed to determine the risks associated with UV radiation since its induced biological effects depend highly on the wavelength. Furthermore, these measurements are also necessary to monitor the short- and long-term trends of solar UV radiation (Zerefos et al., 2012; Fountoulakis et al., 2016), to test radiative transfer models (Mayer et al., 1997) as well as to validate satellite products (Eck et al., 1995; Kazantzidis et al., 2006; Antón et al., 2010). In addition, they are also used to study the

effect of ozone, clouds and atmospheric aerosols on the irradiance that reaches the Earth's surface (Bernhard et al., 2007; Seckmeyer et al., 2008).

Double monochromator scanning spectroradiometers are the preferred devices to measure UV spectral radiation due to their stray-light reduction and linearity. However, their high economic cost, slow scanning, difficulties to transport and demanding maintenance limit their large-scale deployment. In this framework, the new cost-effective spectroradiometers,

based on arrays of CCD sensors, appear as an interesting alternative because of their fast scanning and compact design. However, as CCD-array spectroradiometers are single monochromators, they are significantly affected by stray light. Consequently, they require either mathematical (Zong et al., 2006; Nevas et al., 2014) or experimental-based (Jäkel et al., 2007; Shaw and Goodman, 2008) corrections to provide accurate solar UV measurements. Furthermore, the array detectors have low sensitivity (Edwards and Monks, 2003; Jäkel et al., 2007), resulting in a higher detection threshold. To improve

their performance, new guidelines and techniques have been developed within several research projects such as the EMRP project ENVO3 "Traceability for surface spectral solar ultraviolet radiation" (Blumthaler et al., 2013; Nevas et al., 2014; Egli et al., 2016) and the EMRP ENV59 "Traceability for atmospheric total column ozone" (Gröbner et al., 2017; Sildoja et al., 2018; Vaskuri et al., 2018).

To overcome the aforementioned challenges, several manufacturers are devoting a considerable effort to the development of

improved instrumentation. In particular, Gigahertz-Optik GmbH has developed the BTS2048-UV-S series CCD-array spectroradiometers (from now on called BTS). Thanks to a hardware-based stray-light correction and a BiTec-Sensor, it measures spectral UV irradiance with good linearity and stray-light reduction (Zuber et al., 2018a).

Several studies have been carried out to assess the quality of the BTS series. Its performance, regarding total ozone column values, is comparable to that provided by Dobson and Brewer instruments (Zuber et al., 2018a, 2021). As for the UV index,

the values derived from the BTS spectra were within ±1% for solar zenith angle (SZA) smaller than 70° in reference to a scanning DTMc300 double monochromator (Zuber et al., 2018b). Additionally, the BTS can measure both direct and global spectral irradiance with a similar quality to that obtained by the double monochromator QASUME (Quality Assurance of Spectral Solar UV Measurements in Europe) and a scanning DTMc300 double monochromator, respectively (Zuber et al., 2018a, b).



Nonetheless, in these previous works, only the short-term performance of the BTS concerning global UV spectral irradiance has been studied. Hence, the range of SZA and UV index covered was narrow, limiting the complete evaluation of the stability and dynamic range of the BTS spectroradiometer. Furthermore, since the BTS has been characterized during short-term comparison campaigns, its seasonal behavior has yet to be evaluated.

Thus, the original contribution of this paper is the study of the long-term performance of the BTS regarding global UV
spectral irradiance. The study also analyzes the diurnal and seasonal dependence of the calibration as well as the performance of the BTS measuring the UV index. The results obtained contribute highly to quantifying the quality of the BTS measurements.

The paper is organized as follows. The characteristics of the spectrometers Brewer #150 and BTS used in this work are described in Section 2. Next, section 3 presents the methodology applied to compare the spectral irradiance of both
instruments. In section 4 the spectral irradiance and UV index ratio (BTS/Brewer) are analyzed. Finally, section 5 summarizes the main conclusions.

## 2 Instrumentation

The spectrometers Brewer #150 and BTS2048-UV-S-WP used in this study are installed at the El Arenosillo Atmospheric Sounding Station, located in Mazagón, Huelva (Spain). It belongs to the Earth Observation, Remote Sensing and
Atmosphere Department of the National Institute of Aerospace Technology (INTA). Every two years, it hosts the Regional Brewer Calibration Center – Europe (RBCC-E) intercomparison campaigns, where Brewers are calibrated for total ozone column (TOC) and global UV irradiance.

### 2.1 Brewer #150

The Brewer MK-III #150 is a double monochromator spectrophotometer (double Brewer) that measures global UV spectral
irradiance between 290 and 363 nm with a step of 0.5 nm. It has a full width half maximum (FWHM) of 0.6 nm and a wavelength accuracy of 0.05 nm. In this configuration, a complete scan takes approximately 4.5 minutes. Instead of the traditional design (a standard flat diffuser), the Brewer #150 features a CMS-Schreder entrance optic (Model UV-J1015) which improves the angular response, reproducibility and accuracy of global irradiance measurements. In fact, its cosine error f2 deviates by 2.4 % from the ideal response, with an uncertainty of 1 % (Gröbner, 2003).

The spectroradiometer is calibrated every two years for solar UV irradiance against the European traveling reference QASUME B5503 (Hülsen et al., 2016), following the methodology set by the Physikalisch-Meteorologisches Observatorium Davos, World Radiation Center (https://projects.pmodwrc.ch/qasume/qasume_audit/reports/). Additionally, it is periodically calibrated with several quartz-halogen standard lamps (1000 W DXW type). Thanks to these calibrations, the quality and accuracy of the UV spectral irradiance measured by the Brewer #150 are guaranteed.





## 2.2 BTS2048-UV-S-WP

The BTS2048-UV-S-WP is a CCD-array spectroradiometer, manufactured by Gigahertz-Optik GmbH. One of his most important features is its BiTec-Sensor (BTS), which combines the properties of an integral detector with those of a spectral detector, resulting in high-end light measurements.

The spectral detector is a spectrometer based on a cooled back-thinned CCD detector with 2048 pixels and an electronic shutter (Zuber et al., 2018a, b). It exhibits a FWHM of 0.8 nm, a pixel resolution of 0.13 nm/pixel and a spectral range of 190 nm to 430 nm. On the other hand, the integral detector consists of a silicon carbide (SiC) photodiode with measurement time ranging from 0.1 to 6000 ms. Since the spectroradiometer is designed for outdoor measurements, it is contained in a weather-proof housing which removes humidity and controls temperature to 38 °C. Regarding the input optics, the BTS2048-UV-S-WP features a cosine corrected diffuser window to improve its angular response, sensitivity and calibration stability.

To overcome the issues most array spectroradiometers face due to the internal stray-light, the BTS spectroradiometer is equipped with several optical filters mounted on a remote-controlled filter wheel (described in detail by Zuber et al., 2018a), ruling out the need for mathematical stray-light correction methods.

## 3 Methodology

To validate the BTS' long-term performance three campaigns measuring global UV spectral irradiance were carried out at the El Arenosillo Atmospheric Sounding Station. The first one was performed from 26 May 2020 to 16 June 2020 (spring 2020), the second one from 05 July 2021 to 15 July 2021 (summer 2021) and the third one from 10 November 2021 to 25 November 2021 (autumn 2021).

During the three campaigns different atmospheric conditions were observed, with cloud-free, partly and totally overcast skies conditions being covered. However, only cloud-free conditions have been considered since the long time required by a scanning spectroradiometer for completing a full scan prevents its use in comparison under varying cloudy conditions. As for the ozone, throughout the spring 2020 campaign it varied from 290 to 333 DU, during the summer 2021 campaign from 284 to 324 DU and in the autumn 2021 it fluctuated between 280 and 325 DU. Regarding the solar zenith angle coverage, the minimum SZA reached was 13.8°, 14.5° and 54.4° during the spring 2020, summer 2021 and autumn 2021 campaigns, respectively. Finally, the UV index ranged from 5.4 to 10.6 in the spring 2020 campaign, from 8.5 to 10.5 through the summer 2021 and from 2.4 to 3.3 during the autumn 2021.

To compare the data registered by both instruments, the measured spectra had to be previously synchronized in time. As mentioned above, the BTS is able to record a full scan within seconds (one timestamp for one complete spectrum) whereas the Brewer takes about 4.5 minutes (a timestamp for each wavelength scanned). To synchronize the scans, only the BTS spectra within ±1 minute of the Brewer's central wavelength (326.5 nm) timestamp have been considered. To limit the amount of data obtained during the campaigns the BTS and Brewer were scheduled to measure every 2 and 15 minutes





respectively. Putting the former criteria into practice resulted in 350, 219 and 142 simultaneous UV spectra for the spring 2020, summer 2021 and autumn 2021 campaigns, respectively.

Finally, since both instruments have different optical bandwidths (FWHM), the measured spectra were first deconvolved with their individual slit function, and then convolved with a 1 nm triangular bandpass using the SHICRivm software package V. 3.075. This methodology also corrects the wavelength shift of the two instruments, with an accuracy of 0.02 nm (Slaper et al., 1995).

## 4 Results

### 4.1 Spectral analysis

To assess the long-term spectral performance of the BTS, the spectral ratios between the synchronized irradiance measurements of the BTS and the Brewer #150 reference are obtained for each measurement campaign. The data covers all SZA lower than 70° and only spectra measured under cloud-free conditions are considered. The average spectral ratio between the BTS and the Brewer #150 for the wavelength range from 290 to 360 nm is shown in Fig. 1 for the spring 2020, summer 2021 and autumn 2021 comparison campaigns.

It can be seen from Fig. 1 that the spectral ratio displays a similar behavior during the three comparison campaigns. The BTS shows a steady underestimation of global irradiance of about –7 % between 310 and 360 nm. As for the other wavelength regions, the spectral ratio decreases between 300 and 310 nm. At shorter wavelengths, below 300 nm, the ratio increases rapidly and deviates by more than 20 %, even though the BTS has a hardware-based correction for stray light. This wavelength threshold of reliable recording, 300 nm, is similar to the one other stray-light-corrected CCD-array spectroradiometers have (Ylianttila et al., 2005; Ansko et al., 2008; Kouremeti et al., 2008; Egli et al., 2016). Overall, the agreement between the two instruments is satisfactory between 300 and 360 nm, as the spectral ratio varies within 5 %.

For each campaign, the mean and variability of the spectral ratio are given in Table 1, separately for the three observed wavelength regions in Fig. 1. Table 1 confirms the previous statement: the two instruments agree within 5 % between 300 and 360 nm. On the other hand, the 290–300 nm region has the largest variability. This was expected since in this wavelength range, the spectral ratio varies abruptly.

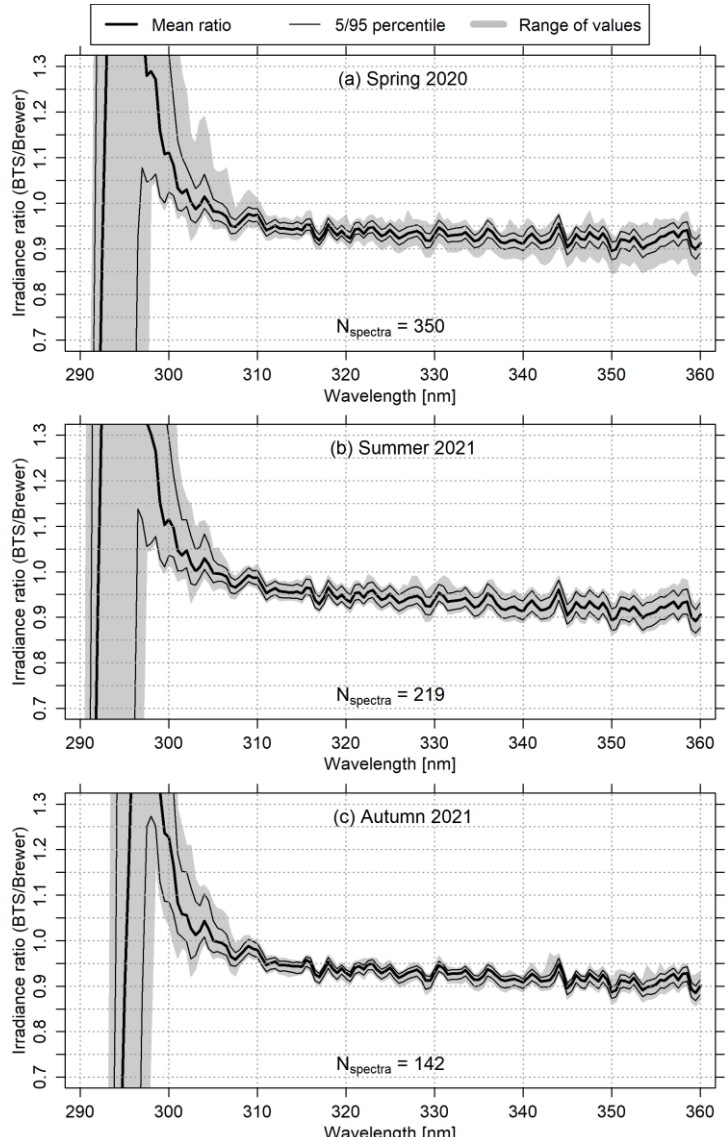

**Figure 1: Average ratios, range of values and 5th/95th percentile of global UV spectral measurements, from cloud-free conditions, between the BTS and the Brewer #150 during (a) spring 2020 (from 26 May to 16 June 2020); (b) summer 2021 (from 5 July to 15 July 2021); and (c) autumn 2021 (from 10 November to 25 November 2021).**

It should be noted that the average ratio, exclusively in the 290–300 nm region, is significantly lower for the autumn 2021 campaign (see Fig. 1). The measurement settings used in all three campaigns were the same and the reason for these large differences is unknown. We may speculate that the BTS' noise filter went below its default level after two years without any calibration checks or recalibration. Hence, all values at this new level are set to 0 and consequently, the ratio plunges (see Table 1). The former statement is also supported by the BTS' manufacturer (personal communication).




Table 1: Summary statistics of the three measurement campaigns with the BTS spectroradiometer relative to the double Brewer spectrometer. The variability is defined as the difference between the 5th and the 95th percentile of all scans.

| Campaign | Number of scans | 290–300 nm | | 300–310 nm | | 310–360 nm | |
| --- | --- | --- | --- | --- | --- | --- | --- |
| | | Mean ratio | Variability (%) | Mean ratio | Variability (%) | Mean ratio | Variability (%) |
| Spring 2020 | 350 | 1.10 | 192.9 | 1.00 | 7.0 | 0.93 | 3.3 |
| Summer 2021 | 219 | 1.18 | 175.4 | 1.01 | 7.2 | 0.93 | 4.4 |
| Autumn 2021 | 142 | 0.81 | 174.9 | 1.02 | 9.6 | 0.93 | 2.7 |

To check the BTS' stability the average ratios between the BTS and the Brewer #150 for the three comparison campaigns are

represented together in Fig. 2. Except for the aforementioned differences observed at short wavelengths (below 297 nm), the ratios during the three campaigns are virtually identical. Therefore, the BTS' calibration was stable during the whole period of study (more than 1 year), despite the fact that no calibration checks were performed during this time. Furthermore, the BTS shows no seasonal dependence.

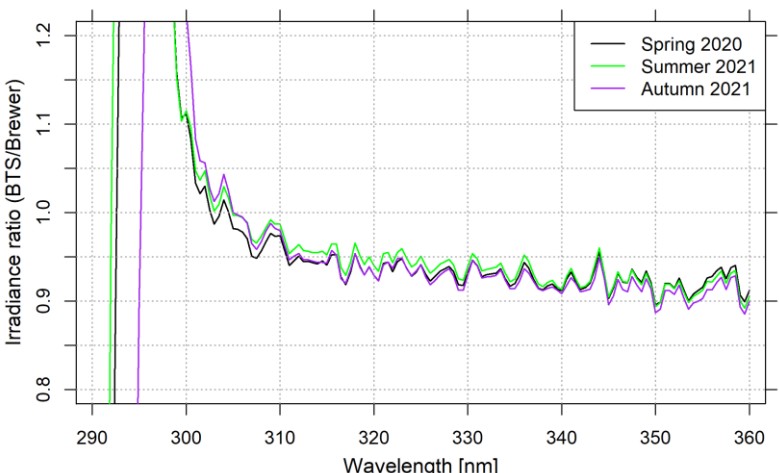

Figure 2: Average spectral ratios obtained throughout the three campaigns.

The spectral ratios in Fig. 1 and 2 are averages of all spectra with sufficient synchronization in time, and as a result, may be biased by systematic diurnal variations. To further describe the performance of the BTS, the ratios between the BTS and Brewer #150 are shown in Fig. 3 for different wavelength bands with respect to SZA. The ratios are averaged in ±2.5 nm wavelength bands at 305, 310, 320 and 350 nm. Wavelengths below 300 nm were not considered since at this wavelength

region the ratio increases sharply (see Fig. 1 and 2).

Figure 3 shows that the spectral ratio at 305 nm has a strong dependence on SZA, increasing with growing SZA. This indicates that, at this wavelength region, spectral UV irradiance should be measured at small SZAs to achieve a better similarity to Brewer #150. At longer wavelengths, over 310 nm, the ratios are very stable, to within less than 10 % and close





to unity. In fact, the BTS shows no diurnal variation in none of the measurement campaigns. As expected, the spectral ratio
behaves identically in all three campaigns, confirming that the BTS shows no seasonal behavior.

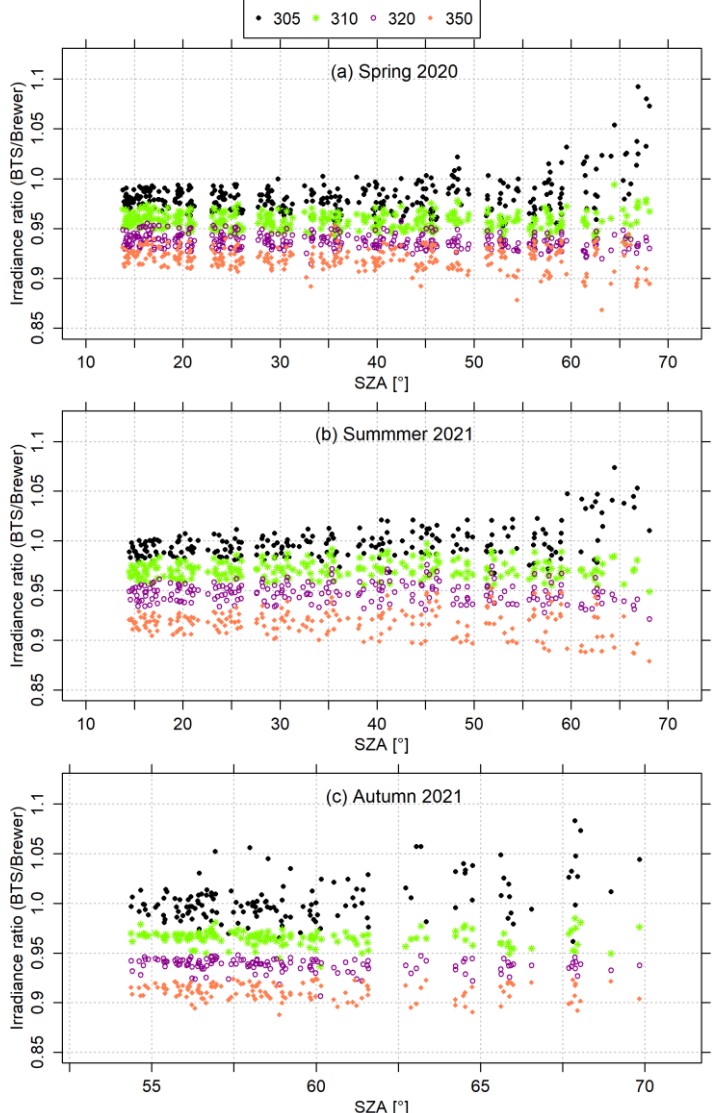

**Figure 3: The ratios of global UV spectral irradiance at selected wavelengths between the BTS and the Brewer #150. The measurements were obtained from cloud-free conditions and SZAs below 70° during (a) spring 2020, (b) summer 2021 and (c) autumn 2021. Each data point is calculated from the average over a ±2.5 nm wavelength band.**

## 4.1 UV index

To evaluate the dynamic range of the BTS, an integrated quantity such as the UV index is analyzed for SZAs less than 70°.

Figure 4 represents, as a function of SZA, the daily variation of the ratios between the UV index measured by the BTS and

the Brewer #150 for the three measurement campaigns. The figure reveals that the ratio is very stable and close to unity.



Overall, the BTS slightly underestimates the UV index, with an average bias of less than 2 % for SZAs below 70°. However,

one should note that this bias is higher for the autumn 2021 campaign, less than 3%, arising from the fall of the spectral ratio

between 290 and 300 nm.

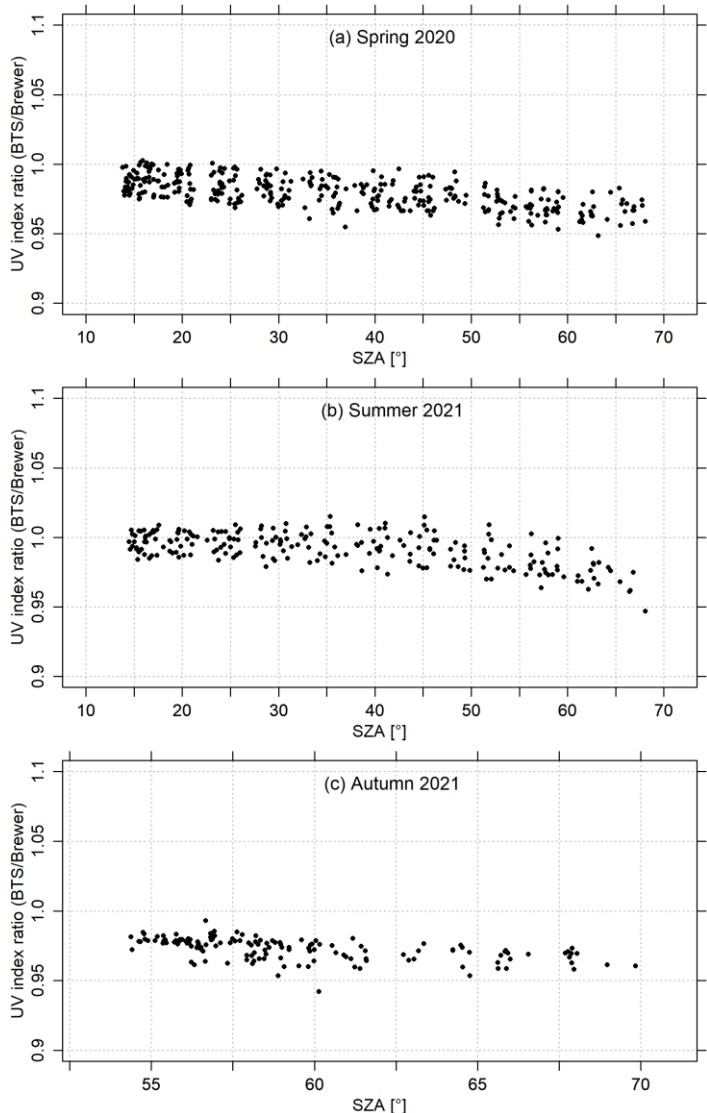

**Figure 4: The ratio of UV indices between the BTS and the Brewer #150, as a function of solar zenith angle. The measurements were obtained from cloud-free conditions and SZAs lower than 70° during (a) spring 2020, (b) summer 2021 and (c) autumn 2021.**

Finally, the UV index values derived from the BTS are compared to the values obtained from the Brewer #150 (see Fig. 5).

A clear linear relationship between the two instruments is found for the UV index, with a coefficient of determination close

to unity. Furthermore, the slope is close to unity, $(0.9953 \pm 0.0013)$, and the intercept is close to zero, $(-0.047 \pm 0.007)$. This



confirms that the BTS underestimates marginally the UV index and that its dynamic range is comparable to that of the Brewer #150.

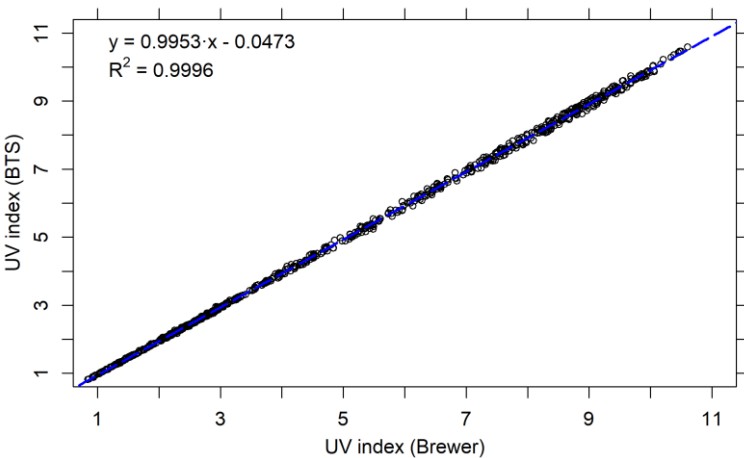

**Figure 5: Synchronized UV index obtained from the BTS versus the ones from the Brewer #150. The measurements were derived from cloud-free conditions and SZAs below 70° combining all the available data of the three measurement campaigns (711 pairs of UVI values).**

## 5 Conclusions

The BTS2048-UV-S-WP long-term performance, regarding global UV spectral irradiance, has been studied via three measurement campaigns compared to a reference such as the double spectroradiometer Brewer #150.

Evaluations of the spectral ratios between the BTS and the Brewer #150 showed that the agreement between the two instruments is satisfactory between 300 and 360 nm, as the spectral ratio is constant, at around 0.94, and varies within 5 %. At shorter wavelengths, below 300 nm, the BTS is unable to detect UV radiation with the same quality as the Brewer #150 due to remaining stray light. This highlights the limitations of the BTS array spectroradiometer to accurately measure the entire UV-B (290–315 nm) range. Furthermore, the comparison of the three average ratios BTS/Brewer obtained throughout each campaign reveals that the BTS has a stable calibration as well as no seasonal behavior. However, calibration checks or recalibrations are advised to ensure the correct functioning of the instrument.

On the other hand, the analysis of the spectral ratios' variation illustrates a marked dependence on SZA for wavelengths shorter than 305 nm. At longer wavelengths, no significant dependence on SZA is found. The ratios were stable, to within less than 10 % and close to unity. Thus, solar UV measurements from the BTS and Brewer #150 spectroradiometers are very consistent.

As for the UV index, the BTS slightly underestimates this integrated quantity, with an average bias of less than 3 % for all SZAs below 70°. Therefore, the BTS is able to provide reliable measurements of the UV index, an important parameter to inform the public about the impact UV has on human health. Moreover, the BTS' bias could be further improved with



specific calibrations. Regarding the comparison between the UV index values measured by the BTS and the Brewer #150, it showed that the dynamic range of the BTS is similar to that of the Brewer #150.

These evaluations confirmed that the BTS' long-term performance of global UV spectral measurements, with its default calibration, has a quality comparable to that provided by a double-monochromator Brewer spectrophotometer in the 300–
360 nm region. Additionally, this study highlights the necessity of intercomparison campaigns to assess the performance of array spectroradiometers. Furthermore, it also shows the importance of repeated site comparisons to evaluate the quality of long-term UV monitoring, calibration's stability, seasonal dependence and dynamic range of the spectroradiometer under study.  Once their quality is assessed, array spectroradiometers could contribute to the enlargement of worldwide solar UV monitoring networks.

*Code and data availability.* The data and code used in this study will be provided after personal communication with the authors of the presented paper.

*Author Contributions.* CG prepared the manuscript with contributions from all co-authors, developed the code and analyzed the data as part of her doctoral thesis. JAB installed the BTS and assisted in its configuration and data acquisition. JMV and AS participated in the conceptualization and provided valuable feedback on the data analysis as well as the writing of the
paper.

*Competing interests.*  The authors declare that they have no conflict of interest**.**

*Acknowledgements.* This work is part of the R+D+i grants RTI 2018-097332-B-C21 and RTI 2018-097332-B-C22 funded by MCIN/AEI/ 10.13039/501100011033/ and "ERDF A Way of Doing Europe", and projects GR18097 and IB18092 funded by Junta de Extremadura and "ERDF A Way of Doing Europe".

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
