# Peer review of "Comparison of global UV irradiance spectral measurements between a BTS CCD-array and a Brewer spectroradiometer"

_Atmospheric Measurement Techniques, 2022_

## Author Comment (AC1)

**Response to Anonymous Referee #1**

Authors' response to Referee #1 comments on "Comparison of global UV irradiance measurements between a BTS CCD-array and a Brewer spectroradiometers".

The authors thank the Referee for the careful and constructive examination of the manuscript and reply to all comments below. The answer is structured as follows: the comments from Referee #1 are marked in red and the authors' response and changes in manuscript are written in black.

This is a very interesting paper on the performance of the BTS CCD-array spectroradiometer, compared with a Brewer spectroradiometer.

My major comments are related:

a. With the explenation of the differences / changes of this ratio with wavelength (mainly)

b. With a conclusion of the measurement uncertainty and accuracy of the Brewer instrument and through this comparison reporting on the uncertainty of the new BTS CCD array

Speciffic comments

Line 39 maybe also include an OMI related validation publication: Arola et al., A new approach to correct for absorbing aerosols in OMI UV DOI: 10.1029/2009GL041137

The reference (Arola et al., 2009) has been added to the manuscript.

Line 62 Probably a reference to the Qasume: Quality assurance of spectral ultraviolet measurements in Europe through the development of a transportable unit (QASUME) DOI: 10.1117/12.468641

The reference (Bais et al., 2003) has been added to the manuscript.

Line 115 probably a table mentioning the dates, names, ozone, temperature , cloud comments of the 3 periods could be useful

The following table has been added in the section "Methodology" to summarize the dates, names, ozone, temperature and number of cloudless spectra of each campaign.

| Campaign | Date | Number of spectra | Temperature (°C) | | Ozone (DU) | |
|---|---|---|---|---|---|---|
| | | | Range | Mean | Range | Mean |
| Spring 2020 | 26/05–16/06 | 350 | 12–28 | 20 | 290–333 | 313 |
| Summer 2021 | 05/07–15/07 | 219 | 16–33 | 23 | 284–324 | 301 |
| Autumn 2021 | 10/11–25/11 | 142 | 5–23 | 13 | 280–325 | 303 |

Line 137 : why have you put the limits for 70 degrees and the cloudless sky ?

The study is restricted to 70⁰ in order to avoid possible issues related to the cosine error, whose contribution can be significant at large solar zenith angles.

Additionally, only cloudless sky conditions have been considered in order to obtain a reliable comparison BTS - Brewer. The Brewer takes 4.5 minutes to measure one spectrum and, therefore, stable conditions are required to obtain a scan that can be compared to the almost instantaneous spectrum measured by the BTS. These required stable conditions are only guaranteed under cloudless conditions.

The previous information has been added to the methodology section as follows:

"Only cloud-free conditions have been considered in order to reliably compare the almost instantaneous spectrum measured by the BTS to the slow-scanned spectrum of the Brewer. Furthermore, the comparison has also been limited to SZAs lower than 70° to avoid possible issues related to the cosine error, whose contribution can be significant at large SZAs."

Line 145: Kouremeti

The reference has been corrected so the author's name was properly written.

Table 1: Variability: is this 1 sigma ?

No, the variability was obtained calculating the difference between the 5th and 95th percentile of all scans. This information has been added to the manuscript, for greater clarity.

Section 4.1:

-Is there any idea for the low but obvious drop of the ratio going from 305 to the end of the spectrum (e.g. fig. 3) ?

Figure 1 shows that the spectral ratio behaves differently depending on the wavelength. In this way, it increases rapidly below 300 nm, while it decreases steadily for longer wavelengths. Consequently, as we move from 305 nm to the end of the spectrum the spectral ratio falls gradually. These differences may be partly due to stray light, calibration procedures and cosine response. This fact has been discussed in the text as follows:

"As expected, the spectral ratio slightly decreases as wavelength increases, displaying the same behavior shown in Figure 1. These differences may be partly due to remaining stray light, cosine response and the different calibration sources for the two instruments."

-Can you comment on the signal to noise ratio for low wavelengths and high solar zenith angles ?

Signal-to-noise ratio decreases as solar zenith angle rises, since the incident radiation traverses a larger path through the atmosphere, increasing its absorption and scattering. This effect is more pronounced for short wavelengths as the signal measured in this wavelength

interval is especially low, according to the spectral distribution of the solar spectrum. This information has been added to the text as follows:

"Signal-to-noise ratio is especially low for short wavelengths according to the spectral distribution of the solar spectrum. This decrease is particularly strong for high SZAs since the radiation is attenuated as it traverses a larger path through the atmosphere".

-Is the curvature of the ratios in figure 2 due to the instrument calibration principles/sources ? or is there any other reason involved?

The curvature could be due to several factors:

1. Calibration sources. The calibration procedure for the Brewer and the BTS are quite different. Brewer #150 is usually calibrated by comparison with the QASUME and 1000 W lamps, while the BTS is calibrated using 250W and 30 W lamps. Furthermore, the setup used to calibrate both instruments also differ, leading to distinct uncertainty sources.
2. Cosine error. Even if the two instruments have improved diffusers the contribution of their angular error cannot be completely neglected.
3. Stray light. Although the BTS has a filter wheel to automatically remove stray light, its spectra are still affected by it, especially at low wavelengths.
4. Ratio's sensitivity. The ratio is very sensitive to small variations, further contributing to the curvature.

All the previous points have been included in the manuscript as follows:

"The curvature observed in Figure 2 could be produced due to several factors such as calibration sources, cosine error, stray light or the ratio's sensitivity to small variations.

Figure 4c: ratios seem slightly lower than the other periods.

Figure 4 has been modified after incorporating the suggestions made by the Anonymous Referee #2. Now, the three charts have the same x-scale and it can be seen that the UV index ratios show the same behavior in the three campaigns. Therefore, ratios may have seemed slightly lower in Figure 4c because of the former x-scale used.

[Figure]

Could you provide an estimation of the Brewer accuracy and uncertainty on deriving UV Index and based on this work to report also on the accuracy and uncertainty of the new instrument ?

An uncertainty estimation of the CCD-array instrument would be very useful for this work.

Following the recommendation of the reviewer, we have attempted to estimate the BTS' uncertainty on deriving the UV Index. Assuming uncorrelated error sources as well as linear effects on the irradiance values, the BTS' uncertainty, regarding the UV index, is $\pm 10\%$.

Nevertheless, this estimation is likely inaccurate since the irradiance measured by a spectrometer is obtained from prior information acquired during the absolute calibration. As a result, errors produced in these previous procedures affect the irradiance data in a nonlinear manner. Furthermore, when there are uncertainty contributions that have a similar effect on the whole spectral measurement, correlations with respect to wavelength may arise.

Therefore, the uncertainty analysis must take into consideration both the nonlinear effects and the possible correlations in the spectral data.

Additionally, to provide a reliable uncertainty estimation a very careful quantification of error sources is needed.

For the Brewer, the uncertainty sources are well-known but hard to assess: radiometric calibration, stray light, linearity, angular response, temperature dependence, wavelength shift and radiometric stability. Thus, the Brewer's uncertainty, regarding spectral UV irradiance, is yet to be accurately determined, even though the Brewers have supported the scientific community for more than 30 years. In fact, as far as we are aware, only the study of Garane et al. (2006) has attempted to estimate it. They, found that the expanded uncertainty of their double Brewer was $\pm 10\%$ following the methodology set by Bernhard and Seckmeyer (1999).

As for the BTS, its global spectral irradiance uncertainty has not been studied either. Currently, only its direct irradiance uncertainty has been assessed. Vaskuri et al. (2018), using a Monte Carlo approach, found it to be $\pm 2.5\%$. However, this value does not completely quantify the BTS uncertainty, since it does not include the angular response, which is one of the most important uncertainty sources. Therefore, reliably estimating the uncertainty of the BTS spectra is a very complex task which is far beyond the scope of this paper.

---

## Author Comment (AC2)

**Response to Anonymous Referee #2**

Authors' response to Referee #2 comments on "Comparison of global UV irradiance measurements between a BTS CCD-array and a Brewer spectroradiometers".

The authors thank the Referee for the careful and constructive examination of the manuscript and reply to all comments below. The answer is structured as follows: the comments from Referee #2 are marked in red and the authors' response and changes in manuscript are written in black.

The paper assesses the performance of a CCD-based spectrometer (BTS) in measuring the spectral solar irradiance in the UV range using data from 3 campaigns that took place in a period of 1.5 years. The comparison was done against data of a double monochromator Brewer spectroradiometer operating regularly at the campaign site. The BTS spectrometer is a rather new instrument and such studies to assess its long term performance are useful contributions for solar UV monitoring. The paper is well structured and addresses most of the usual aspects of intercomparisons of radiation instruments. To my opinion it is in a good stage to be accepted for publication, but I believe with some extra work as suggested in my specific comments below, the results could be further improved and possibly better substantiated. The language of the paper in good, despite some small flaws; some of them are mentioned in the "technical comments" section below.

Specific comments

1, 1: In the title, I suggest adding the word "spectral" before measurements.

The word "spectral" has been added to the title.

3, 66: You actually mean range of intensity, therefore I suggest to avoid using the term UV Index in this context.

The term "UV index" has been replaced by "intensity".

4, 98: What is "high-end light measurements"?

The text "high-end light" has been replaced by "high quality", for greater clarity.

4, 101: The "measurement time ranging from 0.1 to 6000 ms" applies only to the photodiode or also to the CCD?

According to the BTS' manual this measurement time only applies to the photodiode. The CCD has an integration time that ranges from 2 µs to 60s. This information has been added to the manuscript.

5, 138: I suggest to draw a darker horizontal line at 1.0, to guide the eye of the reader and make the comparison amongst the three panels easier. This applies also to figures 3 and 4.

Figures 1, 2, 3 and 4 have been modified to include a horizontal line at 1.0.

5, 143: Concerning the increasing ratio towards shorter wavelengths below 300 nm, this could be partly produced by the cosine response of the BTS diffuser, if the cosine error is larger than the Brewer's. Please include this information in section 2.2, and if the error is larger than the Brewer's I suggest including a brief discussion. Moreover, from figure 1, I don't think that the 5% agreement is valid down to 300 nm. I would be more conservative to the lower limit (e.g. closer to 305 nm). This is also evident from table 1, where only the last column shows variabilities below 5%, contradicting the statement of line 148.

Following the reviewer's comment, information about the cosine response of the Brewer and the BTS spectrometers has been included in sections 2.1 and 2.2, respectively. Since both instruments are equipped with improved diffusers, the difference in the cosine response does not completely explain the increasing ratio towards shorter wavelengths below 300 nm. An additional source of discrepancy could be the stray-light, which becomes more significant as the wavelength decreases. Although both instruments are equipped with means to reduce the stray-light, its contribution cannot be totally neglected. Thus, this information has been included in the manuscript as follows:

"This increase in the ratio could be partly due to stray light and cosine response. Although both instruments are equipped with improved diffusers and stray-light reduction, their contribution cannot be totally neglected."

Moreover, we agree that the more conservative 305 nm is a more suitable threshold and the text has been modified accordingly.

6, 157-159: The discussion around the noise level and its reset to 0 is not clear for inexperienced readers.

For greater clarity, the discussion has been rewritten as follows:

"Figure 1 shows that the average ratio is significantly lower for the autumn 2021 campaign exclusively in the 290–300 nm region. This behavior could be likely related to several factors, such as stray light, differences in the detection threshold between Brewer and BTS, and the BTS's noise reduction filter. These factors have a larger effect for low signals, which are more frequent during autumn due to the lower range of solar elevation as compared with the other two campaigns."

7, 173: For the plots of Figure 3, a more stringent time synchronization could be achieved for each wavelength band (of ±2.5 nm) as opposed to the general synchronization based on the time at 326.5 nm. To be clearer, I mean to compare the data based on the difference between the time the central wavelength of each band is measured and the time of the BTS spectrum. This might further improve the results, especially at larger SZAs when small time differences increase notably the irradiance level. Actually, this might explain a small part the deviations at the shorter wavelengths, in addition to stray-light and (possibly) to cosine response.

The comment of the reviewer has been followed and, after synchronizing with respect to each band's central wavelength, better agreement between measurements has been achieved. The number of outliers has decreased and the increase observed at short wavelengths for large

SZAs has been reduced. As the reviewer said, the synchronization was partly responsible for this increase.

The suggested synchronization has been applied, resulting in the following figures:

[Figure]

Furthermore, the Methodology section has been modified to describe the different synchronization methods that have been essayed (UV index and spectral ratio), as follows:

"However, to further improve the results, different synchronization criteria were applied to study the UV index and angular dependence of the BTS. In this way, to obtain the UV index, only the BTS spectra within ±1 minute of the Brewer's 307 nm timestamp have been considered. This wavelength was selected since the erythemally weighted irradiance peaks between 306 and 308 nm, depending on SZA and total ozone. To analyze the angular

dependence, the spectral ratio BTS/Brewer has been calculated in four different wavelength bands. For each band, the ratio was obtained using BTS spectra within ±1 minute of the Brewer central wavelength (305, 310, 320 and 350 nm) of each band."

The Results section has also been modified to reflect that the angular dependence is slight at short wavelengths.

8, 183: At the caption of Figure 3 please add a note to alert the reader for the x-axis scale change in the bottom panel. The same holds for Figure 4.

The x-axis scale of Figures 3 and 4 has been modified to be the same, so as to facilitate the comparison between figures.

8, 185: A different time synchronization could also be applied for the UV Index comparisons, instead of the time at 326.5 nm. As the erythemally weighted irradiance peaks at between 306-308 nm (depending on SZA and total ozone) the time in this wavelength range would be more appropriate for the comparison and I believe would also improve the results.

The comment of the reviewer has been followed and the UV index has been recalculated using spectra synchronized with respect to 307 nm. The new values of UV index ratio show a less marked dependence with SZA but larger scatter. When comparing the UV index measured by both instruments the linear regression is practically the same as the one obtained with the previous synchronization (326.5 nm).

Thus, the suggested synchronization has been applied and the methodology and figures have been changed accordingly. The results obtained are very similar and the conclusions drawn remain valid, confirming the robustness of the study. The new figures follow:

[Figure]

As for the methodology modification, it has already been shown in the comment regarding the spectral ratio dependence with SZA (Line 7, 173).

Technical Comments

2, 45: "on arrays of CCD sensors". Do you mean "on arrays or CCD sensors"? Otherwise, just say "on CCD sensors".

The phrase has been corrected and changed to "on CCD sensors".

2, 54: Replace "a considerable effort" with "considerable efforts"

The term "a considerable effort" has been replaced with "considerable efforts".

3, 70: Replace "calibration" by "sensitivity"

The term "calibration" has been replaced by "sensitivity"

3, 84: Omit the unnecessary term (double Brewer).

The term "double Brewer" has been omitted.

4, 99: "The spectral detector is a spectrometer". This doesn't make sense. Maybe you can omit "a spectrometer"?

The term "spectrometer" has been omitted.

5, 149: Replace "is similar to the one other stray-light-corrected CCD-array spectroradiometers have" with "is similar to other stray-light-corrected CCD-array spectroradiometers"

The phrase has been corrected and changed to "is similar to other stray-light corrected CCD-array spectroradiometers".

10, 200: I would prefer to see Figure 5 with axes of equal length.

Figure 5 has been changed to reflect the new synchronization and to have axes of the same length. The new figure follows:

[Figure]

10, 208: I assume you mean agreement within ±5%.

Exactly. The phrase has been corrected and changed to "the spectral ratio is constant, at around 0.94, and agrees within 5 %."

11, 221: Replace "specific" with "regular"

The term "specific" has been replaced by "regular